

# Retrospective study on structural neuroimaging in first-episode psychosis

Ricardo Coentre[1,2], Amilcar Silva-dos-Santos[3] and Miguel Cotrim Talina[3,4]

[1] First-Episode Psychosis Program, Department of Psychiatry, Hospital Vila Franca de Xira, Vila Franca de Xira, Portugal
[2] Faculty of Medicine, University of Lisbon, Lisbon, Portugal
[3] Department of Psychiatry, Hospital Vila Franca de Xira, Vila Franca de Xira, Portugal
[4] CEDOC, Chronic Diseases Research Centre, Nova Medical School, Lisbon, Portugal

## ABSTRACT

**Background.** No consensus between guidelines exists regarding neuroimaging in first-episode psychosis. The purpose of this study is to assess anomalies found in structural neuroimaging exams (brain computed tomography (CT) and magnetic resonance imaging (MRI)) in the initial medical work-up of patients presenting first-episode psychosis.

**Methods.** The study subjects were 32 patients aged 18–48 years (mean age: 29.6 years), consecutively admitted with first-episode psychosis diagnosis. Socio-demographic and clinical data and neuroimaging exams (CT and MRI) were retrospectively studied. Diagnostic assessments were made using the Operational Criteria Checklist +. Neuroimaging images (CT and MRI) and respective reports were analysed by an experienced consultant psychiatrist.

**Results.** None of the patients had abnormalities in neuroimaging exams responsible for psychotic symptoms. Thirty-seven percent of patients had incidental brain findings not causally related to the psychosis (brain atrophy, arachnoid cyst, asymmetric lateral ventricles, dilated lateral ventricles, plagiocephaly and *falx cerebri* calcification). No further medical referral was needed for any of these patients. No significant differences regarding gender, age, diagnosis, duration of untreated psychosis, in-stay and *cannabis* use were found between patients who had neuroimaging abnormalities *versus* those without.

**Discussion.** This study suggests that structural neuroimaging exams reveal scarce abnormalities in young patients with first-episode psychosis. Structural neuroimaging is especially useful in first-episode psychosis patients with neurological symptoms, atypical clinical picture and old age.

# INTRODUCTION

Frequently, structural neuroimaging exams (brain computed tomography (CT) and magnetic resonance imaging (MRI)) are requested as part of the initial medical work-up in young patients with first-episode psychosis (*Freudenreich, Schulz & Goff*, *2009*). The aim is to exclude "organic" brain causes that could explain psychotic symptoms and modify

Corresponding author
Ricardo Coentre,
Ricardo.Coentre@netc.pt

**Table 1  Some current guidelines regarding neuroimaging in first-episode psychosis.**

| National Institute for Health and Clinical Excellence (2008) | American Psychiatric Association (2006) | Canadian Psychiatric Association (2005) | Royal Australian and New Zealand College of Psychiatrists 2005 (McGorry, 2005) |
|---|---|---|---|
| "Structural neuroimaging techniques (either magnetic resonance imaging (MRI) or computed axial tomography (CT) scanning) are not recommended as a routine part of the initial investigations for the management of first-episode psychosis." | "A CT or MRI scan may provide helpful information, particularly in assessing patients with a new onset of psychosis or with an atypical clinical presentation." | "Excluding patients with head injury, neurologic disease, seizures, or substance abuse, 7.9% of MRI scans obtained in first-episode patients were of "clinical importance, affecting prognosis, diagnosis, or management." | Baseline MRI scan should be done as part of optimal initial assessment in first-episode psychosis. |
| "…this decision should not affect the current practice of using structural neuroimaging techniques selectively to exclude organic causes of psychosis where people's symptoms, or other aspects of their presentation, suggest a higher likelihood of an underlying organic cause." | | "Recommends CT or MRI in baseline assessment of first-episode psychosis." | |

management and treatment (*Woolley*, *2005*). These include brain injury, demyelinating disease, tumours, multiple sclerosis or stroke. In recent years, much research has been made in both structural and functional neuroimaging in early phases of psychosis (*Dazzan et al.*, *2015*; *Hager & Keshavan*, *2015*; *Jardri*, *2013*; *Jung et al.*, *2010*; *Strakowski et al.*, *2008*; *Tognin et al.*, *2014*). However, the impact of these studies in clinical practice has been disappointing.

Guidelines are divergent regarding neuroimaging in first-episode psychosis. Table 1 summarises some of the current major guidelines (*American Psychiatric Association*, *2006*; *Canadian Psychiatric Association*, *2005*; *McGorry*, *2005*; *National Institute for Health and Clinical Excellence*, *2008*). When admitting young patients with first-episode psychosis with no positive neurological findings, physicians are faced with the decision if brain imaging should be performed. Early identification of "organic" lesions responsible for psychotic symptoms could lead to a change in the treatment because the lesions may be surgically or medically treatable. Initial reports seemed promising, with abnormality rates of 30%, but with time most of these findings in structural neuroimaging were incidental and not causal to the psychotic picture (*Goodstein*, *1985*; *Weinberger*, *1984*).

Published studies show rates of overall structural neuroimaging abnormalities between 0% and 65.2% in patients with first-episode psychosis. Incidental unrelated to the psychosis structural neuroimaging abnormalities are found between 2.4% and 65.2% of patients and neuroimaging abnormalities link to psychosis in 0%–2.7% (*Bain*, *1998*; *Battaglia & Spector*, *1988*; *Gewirtz et al.*, *1994*; *Khandanpour, Hoggard & Connolly*, *2013*; *Lubman et al.*, *2002*; *Robert Williams, Yukio Koyanagi & Shigemi Hishinuma*, *2014*). Most of researches only studied CT neuroimaging (*Battaglia & Spector*, *1988*; *Bain*, *1998*; *Gewirtz et al.*, *1994*; *Strahl, Cheung & Stuckey*, *2010*), one included only MRI neuroimaging (*Lubman et al.*, *2002*) and

three studies included CT and MRI neuroimaging (*Goulet et al.*, *2009*; *Khandanpour, Hoggard & Connolly*, *2013*; *Robert Williams, Yukio Koyanagi & Shigemi Hishinuma*, *2014*). Therefore, divergent results in structural neuroimaging abnormalities rates could be mainly explained by different patient age included. As expected, studies where older patients with first-episode psychosis were included more neuroimaging abnormalities were found. For example *Gewirtz et al.* (*1994*) studied first-episode psychosis patients with a mean age of 35, ranging between 18 and 66 years. CT neuroimaging was studied and 42.3% revealed benign and nonspecific abnormalities (diffuse cortical atrophy, arachnoid cysts, ventricular enlargement and venous angioma) and 2.4% of abnormalities link to psychosis (arachnoid cyst in right temporal area, bilateral parietal and subinsular infarcts, bilateral parietal ischemic changes and colloidal cyst in the third ventricule with obstrutction of the foramen of Munro) (*Gewirtz et al.*, *1994*). By the contrary in one published study that included only young first-episode psychosis patients (age between 12 and 30 years) (*Robert Williams, Yukio Koyanagi & Shigemi Hishinuma*, *2014*) revealed 5.2% of incidental neuroimaging findings with none considered to be causal to psychosis. When included only the few studies published with young first-episode psychosis patients, overall non causal structural neuroimaging abnormalities results revealed rates from 2.2% to 13.2%, and abnormalities link to psychosis from 0% to 1.3% (*Bain*, *1998*; *Goulet et al.*, *2009*; *Lubman et al.*, *2002*; *Robert Williams, Yukio Koyanagi & Shigemi Hishinuma*, *2014*).

CT and MRI are the most used structural neuroimaging techniques in daily clinical practice. CT scans are assessable and take little time, but deliver radiation. MRI carries no risk of radiation exposure, provides better spatial resolution and grey-white matter differentiation than CT. MRI is not widely available mainly because of the cost compared with CT. MRI is particularly useful in epilepsy, multiple sclerosis, small brain tumours and vasculitis.

The aim of this study is to evaluate abnormalities found in neuroimaging exams in a sample of young patients with first-episode psychosis. Complementing the few studies published, we try to contribute to more robust evidence-based decisions about structural neuroimaging in clinical practice in young first-episode psychosis patients. We also tried to overcome existing limitations of previous published studies, including only patients with first-episode psychosis diagnosis, young patients (age < 50 years), consecutive patient series and referred all details of study design. We included as well CT and MRI neuroimaging and in- and outpatients approaching to daily clinical routine.

## MATERIALS AND METHODS

### Sample

The sample included consecutive, non-affective, first-episode psychosis patients, presenting to the Department of Psychiatry of Hospital Vila Franca de Xira, Portugal, from August 2013 to September 2015. Hospital Vila Franca de Xira is a general hospital in the metropolitan area of Lisbon, Portugal, with a population of 245,000. Psychosis was defined according DSM-IV criteria (*American Psychiatric Association*, *1994*). Inclusion criteria were an individual experiencing his or her non-affective first-episode psychosis, aged 16–48 years and living in the area of Hospital Vila Franca de Xira. The study was approved by Hospital

Vila Franca de Xira Ethics Committee. In addition to the written informed consent to the brain imaging that all patients give during the daily routine of neuroimaging, each participant gave a specific written informed consent to the present study.

## Data collection

The medical electronic files of the patients were reviewed retrospectively. Socio-demographic and clinical data, neuroimaging images (CT and MRI) and respective reports were analysed by an experienced consultant psychiatrist. CT scans were performed in Siemens SOMATON Emotion 16 CT machine with 2.4 mm section thickness (Siemens Healthcare). All CT studies were considered for contrast administration after review of non-contrasted study. In 12 patients contrast for contrast-enhanced CT scan was administered. MRI were performed in Toshiba Titan 1.5 T machine (Toshiba Medical Systems Europe). All participants evaluated using MRI had sagittal T1-weighted, axial T2-weighted, axial T2-weighted Flair, axial gradient-echo, axial diffusion-weighted and coronal T2-weighted imaging. Gadolinium-enhanced MRI was used in all MRI studies performed (axial, sagittal and T1-weighted coronal sequences). CT and MRI images were analyzed using Impax 6.5. Software (Agfa Healthcare Inc.). All CT and MRI reports had been written by a consultant neuroradiologist. When a doubt persisted after examination of the neuroimaging images and reports, collaboration of a consultant neuroradiologist was used to reanalyse the exams. Brain images (CT and/or MRI) are used in daily practice in the clinical evaluation of all first-episode psychosis presenting to the Department of Psychiatry as part of the initial medical work-up. Patients with only non-brain abnormalities in CT scan or MRI (sinus disease, sebaceous cyst or dermoid cyst) were included in patients without neuroimaging abnormalities. We used the Operational Criteria Checklist + (OPCRIT+) instrument to achieve DSM-IV diagnosis (*McGuffin, Farmer & Harvey*, *1991*; *Rucker et al.*, *2011*). OPCRIT+ is a checklist including items of psychiatric history and psychopathology. Checklist ratings are entered into the OPCRIT+ software, which generates a diagnosis for the main categories of affective and psychotic disorders defined according to major classification systems, including DSM-IV. OPCRIT+ has been shown to have good reliability when used by different raters. The rater was an experienced consultant psychiatrist, trained in the use of OPCRIT+.

## Statistical analysis

Statistical analysis was made using SPSS statistics 21 for Windows (SPSS, Chicago, IL, USA). Descriptive analyses are presented as proportions for count data and as means with standard deviations (SD) for continuous data. We separated the samples in two groups, according to the existence of neuroimaging abnormalities (i.e., with vs. without neuroimaging abnormalities), and compared some socio-demographic and clinical characteristics using chi-squared test for categorical variables (or Fisher exact tests as appropriate) and student's $t$-test for continuous variables. The level of statistical significance was $p < 0.05$.

## RESULTS

There were a total of 32 patients who met the inclusion criteria (male = 59.4%). The mean age was 29.6 years (SD = 8.7), ranging from 18 to 48 years. A total of 65.6% were single,

18.8% married and 15.6% divorced. Twenty-two percent of the participants were students, 53.13% unemployed and 25% employed. Of the sample, 6.3% participants lived alone. There was cannabis use in 53.13% of the participants, and 65.63% had an in-stay during acute phase of the illness. Mean duration of untreated psychosis (DUP) was 76.5 weeks (SD = 107.4). According to DSM-IV diagnosis, 31.25% had the diagnosis of schizophrenia, 40.63% psychotic disorder not otherwise specified, 21.88% cannabis-induced psychotic disorder and 6.25% delusional disorder. All patients were on first days of antipsychotic medication when CT scan or MRI was performed. Regarding the type of medication, 31.25% patients were treated with risperidone oral, 21.88% with olanzapine oral, 15.63% paliperidone palmitate long-acting injectable, 9.38% aripiprazole oral, 9.38% paliperidone oral, 6.25% haloperidol oral, 3.13% risperidone long-acting injectable and 3.13% quetiapine fumarate extended-release oral.

Twenty-nine (90.63%) patients received a CT, 1 (3.13%) an MRI and 2 (6.25%) both CT and MRI. None of the neuroimaging findings was considered to be a potential or significant contributory cause to the psychotic episode. Twelve patients (37.5%) had incidental brain lesions not causally related to the psychosis: brain atrophy ($n = 4$), arachnoid cyst ($n = 3$), asymmetric lateral ventricles ($n = 2$), dilated lateral ventricles ($n = 1$), plagiocephaly ($n = 1$) and *falx cerebri* calcification ($n = 1$).

There were no statistically significant differences between participants groups (with vs. without neuroimaging abnormalities) regarding gender ($p = 0.687$), age ($p = 0.490$), DUP ($p = 0.399$), psychiatric diagnosis ($p = 0.069$), in-stay ($p = 0.582$) and cannabis use ($p = 0.108$) (Table 2).

## DISCUSSION

In our sample, no neuroimaging lesions responsible for psychotic symptoms were found. Minor incidental abnormalities were present in 37.5% of the patients. Despite the relevant methodological differences compared to previous published studies, our results are in line with other studies that included young patients in first-episode psychosis. In 1998, Battaglia & Spector studied 45 patients with first-episode psychosis who received a CT scan, and three scans showed positive findings that correlated with neuropsychiatric symptomatology (*Battaglia & Spector*, *1988*). *Goulet et al.* (*2009*) studied 46 patients with first-episode psychosis, in which 44 had CT scans, two had MRI and only one patient showed a lipoma above the pineal gland with no relation with clinical picture. Like our research, previous published studies show that findings in structural brain imaging in young patients with first-episode psychosis are limited regarding clinical utility.

Contrary to our sample in which only young people with first-episode psychosis were included, the mentioned study published by *Khandanpour, Hoggard & Connolly* (*2013*) included first-episode psychosis ranging between 16 and 95 years, including MRI and CT scans. Among patients who underwent MRI, 2.7% had brain lesions potentially responsible for the psychosis (brain tumour, meningioma and human immunodeficiency virus encephalopathy), and 62.5% had incidental brain lesions. Among patients who had CT scans, 1.5% had focal brain lesions potentially accountable for the psychosis, and 65.2%

**Table 2** Socio-demographic and clinical characteristics of patients with and without neuroimaging abnormalities.

| | Patients with neuroimaging abnormalities $n = 11$ | Patients without neuroimaging abnormalities $n = 21$ | P-value |
|---|---|---|---|
| Gender | | | |
| Male | 6 | 13 | NS |
| Female | 5 | 8 | |
| Age | | | |
| Mean years (SD) | 28.09 (9.40) | 30.38 (8.49) | NS |
| Diagnosis DSM-IV | | | |
| Schizophrenia | 2 | 8 | |
| Delusional disorder | 2 | 0 | NS |
| Cannabis induced Psychotic disorder | 4 | 3 | |
| Psychotic disorder not Otherwise specified | 3 | 10 | |
| DUP | | | |
| Mean weeks (SD) | 53.92 (58.55) | 88.27 (125.48) | NS |
| In-stay (n) | 7 | 14 | NS |
| Cannabis use | 8 | 9 | NS |

Notes.

NS, not significant; SD, standard deviation; DSM-IV, Diagnostic and Statistical of Mental Disorders, fourth edition; DUP, duration of untreated psychosis.

had incidental brain lesions not responsible to the psychosis. These results seem to indicate that in older (>50 years) first-episode psychotic patients, structural neuroimaging exams identify a higher number of organic lesions responsible for the psychosis, even so in low rate.

As expected, we did not find any significant differences between the group of patients with incidental brain findings and those without, reflecting that these neuroimaging findings do not reflect a subgroup of patients with particular socio-demographic or clinical characteristics (e.g., longer DUP or more in-stay) (Table 2).

None of the studies chose patients randomly, including our own. But contrary to most of published studies, we include patients consecutively, reducing selection bias.

Only one study that specifically investigated economical factors of structural neuroimaging in psychosis was found (*Albon et al.*, *2008*). The authors concluded that if screening with structural neuroimaging was implemented in all patients presenting with psychotic symptoms before 65 years, little would be found to affect clinical management. The authors also emphasised that there is a paucity in good-quality evidence on the clinical benefits of structural neuroimaging on which to base economic research; thus, the outcome from an economic perspective is not clear.

Our study has some limitations. First, our sample is limited in size, which can limit generalisation of the findings. However, we think it is representative, and our results are similar to other studies in which larger samples were included. Second, we did not include a control group (patients without first-episode psychosis) and therefore did not compare

prevalence and type of the radiological findings. Third, few patients included had an MRI, but taking into account previous studies, this would likely not significantly alter our findings.

## CONCLUSIONS

In conclusion, our results are concordant with the few previous studies in which clinically relevant abnormalities in structural neuroimaging in young patients with first-episode psychosis are scarce. We think that neurological examination is a valuable adjunct instrument once it can help to determine if a patient has signs of a brain lesion, so structural neuroimaging should be performed. However, sometimes the psychiatric status did not permit the elaboration of a good-quality neurological examination, so it would be desirable to improve clinical status first and after doing a neurological examination. Neuroimaging exams should specially be performed in some clinical pictures, such as in patients with neurological symptoms or signs, atypical clinical picture, delirium/organic suggestive symptoms (visual hallucinations, disorientation, memory loss and blurred conscience) and old age (age > 50 years). When a neuroimaging exam is indicated, an MRI should probably be performed because of previously mentioned advantages. MRI should be especially preferred if epilepsy, multiple sclerosis, small tumours and vasculitis are major diagnostic hypotheses.

### Funding
The authors received no funding for this work.

### Competing Interests
The authors declare there are no competing interests.

### Author Contributions
- Ricardo Coentre conceived and designed the experiments, performed the experiments, analyzed the data, contributed reagents/materials/analysis tools, wrote the paper, prepared figures and/or tables, reviewed drafts of the paper.
- Amilcar Silva-dos-Santos conceived and designed the experiments, analyzed the data, wrote the paper, reviewed drafts of the paper.
- Miguel Cotrim Talina conceived and designed the experiments, analyzed the data, wrote the paper.

### Human Ethics
The following information was supplied relating to ethical approvals (i.e., approving body and any reference numbers):
    Hospital Vila Franca de Xira Ethics Committee.

### Data Availability
    The raw data has been supplied as Data S1.

## Supplemental Information

Supplemental information for this article can be found online at http://dx.doi.org/10.7717/peerj.2069#supplemental-information.

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
