# Peer review of "Retrospective study on structural neuroimaging in first-episode psychosis"

_PeerJ, doi:10.7717/peerj.2069_

## Round 0.1 · original submission · Major Revisions

Dear Authors,

There are important comments from both Peer Reviewers which I hope your team can use to revise and add more information.

From Peer Reviewer 1: Add more literatures about neuroimaging results when there is negative focal neurological features. As the findings are not new and already known;then please expand on what is new about this manuscript?

There are publications that indicate negative imaging results for almost all patients who have no neurological deficit and is this applicable also for psychotic patients without neurological deficits?

From Peer Reviewer 2 : Importantly the definition of psychosis: Why did you use CAARMS criteria to define psychosis? This seems unusual. Surely DSM-IV criteria would be more appropriate here, especially given that you later describe that diagnoses were according to DSM-IV.

Lines 99 ff. Details on how MRI and CT scans were acquired are missing and should be added (i.e. the make and model of scanner, sequences, etc.). Also please add information on how images were viewed (i.e. which software was used to reconstruct and display the images).

Thank you

·

Basic reporting

Literature review, objectives and methods are acceptable and good. However, it need to get more literatures about neuroimaging results when there is negative focal neurological features.

Experimental design

nil

Validity of the findings

valid

Additional comments

<> the whole write up is good. But it is a retrospective record review.
<> the findings are not new and already known.
<>A lots have been published that there will be negative imaging results for almost all patients who have no neurological deficit. This is applicable also for psychotic patients without neurological deficit.

Reviewer 2 ·

Basic reporting

Basic reporting of this manuscript is fine.

Experimental design

The design is appropriate, but some questions remain concerning image data acquisition. See more detailed comments below.

Validity of the findings

The findings appear robust and the conclusions valid.

Additional comments

In this manuscript, Coentre and colleagues present a retrospective report of neuroimaging (CT/MRI) abnormalities in a sample of first-episode psychosis patients. It is investigated how frequent abnormalities in these patients are and what the clinical and demographic associations are. The study makes a contribution to answering the clinical question of what may be expected from neuroimaging in the clinical work-up and subsequent treatment of this patient group.

Minor Comments

Line 70: What is B.K.?

Lines 77/78: What do you mean by “MRI is not yet widely available”? Do you have numbers to back this up?

Line 85: What is a “clearly referred study design”?

Lines 91-93, definition of psychosis: Why did you use CAARMS criteria to define psychosis? This seems unusual. Surely DSM-IV criteria would be more appropriate here, especially given that you later describe that diagnoses were according to DSM-IV.

Lines 99 ff. Details on how MRI and CT scans were acquired are missing and should be added (i.e. the make and model of scanner, sequences, etc.). Also please add information on how images were viewed (i.e. which software was used to reconstruct and display the images).

Line 107/108: This sentence is not clear to me at all. Please rephrase and put more clearly.

Lines 235/273: These references appear incomplete.

---

## Round 0.2 · accepted · Accept

Dear Authors, congratulations on the Acceptance.

·

Basic reporting

Thanks for amending the write up.
Background description is enough.

Experimental design

Simple retrospective study.
Method is clear. Simple analysis.
Acceptable.

Validity of the findings

All were negative results as expected. Acceptable.

Additional comments

Scientific merit is not new even though the study has based on young samples.

Reviewer 2 ·

Basic reporting

No further comments

Experimental design

No further comments

Validity of the findings

No further comments

Additional comments

No further comments